# Revisiting nnU-Net for Iterative Pseudo Labeling and Efficient Sliding Window Inference

Ziyan Huang[1,2,†][0000−0002−1533−5239], Haoyu Wang[1,2,†], Jin Ye[2], Jingqi Niu[1,2], Can Tu[1,2], Yuncheng Yang[1,2], Shiyi Du[2,3], Zhongying Deng[2,4], Lixu Gu[1], and Junjun He[2⋆]

1 Shanghai Jiao Tong University, Shanghai, China
`{ziyanhuang, small_dark}@sjtu.edu.cn`
2 Shanghai AI Lab, Shanghai, China
`hejunjun@pjlab.org.cn`
3 Sichuan University, Sichuan, China
4 University of Surrey, Guildford, United Kingdom

**Abstract.** nnU-Net serves as a good baseline for many medical image segmentation challenges in recent years. It works pretty well for fully-supervised segmentation tasks. However, it is less efficient for inference and cannot effectively make full use of unlabeled data, both of which are vital in real clinical scenarios. To this end, we revisit nnU-Net and find the trade-off between efficiency and accuracy in this framework. Based on the default nnU-Net settings, we design a co-training framework consisting of two strategies to generate high-quality pseudo labels and make efficient inference respectively. Specifically, we first design a resource-intensive nnU-Net to iteratively generate high-quality pseudo labels for unlabeled data. Then we train another light-weight 3D nnU-Net using labeled data and selected unlabeled data, with high-quality pseudo labels used for the latter to achieve efficient segmentation. We conduct experiments on the FLARE22 challenge. Our resource-intensive nnU-Net achieves the mean DSC of 0.9064 on 13 abdominal organ segmentation tasks and ranks first on the validation leaderboard. Our light-weight nnU-Net shows the mean DSC of 0.8773 on the validation leaderboard but it makes a better trade-off between accuracy and efficiency. On the test set, it shows the mean DSC of 0.8864, the mean NSD of 0.9465, and the average inference time of 14.59s and wins the championship of the FLARE22 challenge. Our code is publicly available at https://github.com/Ziyan-Huang/FLARE22.

**Keywords:** Segmentation · Semi-supervised learning · Computational Efficiency

## 1 Introduction

Abdominal organ segmentation is an important prerequisite of many clinical applications. In recent years, deep learning based methods are widely used to

---

⋆ Corresponding Author

segment abdominal organs automatically. One of the most important baselines among these methods is nnU-Net[5] and many top solutions for medical image segmentation challenges in recent years are built based on it. Although nnU-Net can achieve state-of-the-art performance in a fully supervised manner, two distinct issues are observed: (1) the default nnU-Net has quadratic computation complexity to volume shape due to sliding-window inference; (2) the default nnU-Net does not support semi-supervised training. However, both the time budget for model inference and the number of labeled data are limited in real clinical scenarios. So, there is a great need for a framework that can make use of unlabeled data and make efficient inference simultaneously.

The Fast and Low-resource Semi-supervised Abdominal Organ Segmentation Challenge 2022 (FLARE22) is a competition that aims at efficiently segmenting 13 organs in CT images from 20+ medical groups. In addition to evaluating the abdominal organ segmentation accuracy, it also takes model efficiency into consideration. By studying the top methods in FLARE21 [7], we find that although the nnU-Net based method [4] can achieve the best DSC and NSD scores, all the top-5 methods do not use nnU-Net. This is probably due to its high resource consumption and low inference speed, making it only rank ninth. We summarized the main efficiency takeoffs from the winning methods in FLARE21: (1) use a small model and low-resolution images; (2) input whole volume image and use two-stage segmentation. Obviously, nnU-Net can also benefit from a small model and low-resolution input. However, inputting the whole volume images will lose the spacing information of medical images. We thus argue that keeping the spacing information and using the sliding-window strategy in nnU-Net is still necessary. The goal of two-stage segmentation is to first locate the region of interest (ROI) with a small computational cost and then conduct fine segmentation only on the ROI to achieve high efficiency. However, the default sliding window inference strategy in nnU-Net spends too much time on the background area, which heavily increases the inference time, especially in whole-body CT images.

The distinction between FLARE22 and FLARE21 is that challenge this year is the semi-supervised learning (SSL) task. In addition to 50 well-annotated images, 2000 unlabeled images are also provided. This setting is reasonable as the pixel-wise annotation is expensive and laborious especially when each pixel of thousands of CT images needs to be annotated into 13 different abdominal organs. Semi-supervised learning method, as a solution to such a dilemma, can be mainly divided into two types: (1) consistency-regularization-based method; (2) pseudo-label-based method. We pick the pseudo-label-based method and combine it with the nnU-Net framework for its simplicity. To achieve high performance, the quality and reliability of pseudo labels are essential. However, we can hardly achieve both efficiency and accuracy using only one model. Thus, we use an efficient small model for inference, while adopting a large model to generate high-quality pseudo labels to train such a small model.

In this paper, we design a framework consisting of two modified 3D nnU-Net to generate high-quality pseudo labels and make inference efficiently respectively.

Specifically, we design a resource-intensive nnU-Net to iteratively generate high-quality pseudo labels for 2000 unlabeled data. Then we conduct image-level selection based on the stability of different re-training iterations and filter out the pseudo labels that are less reliable. To achieve high inference efficiency, we first train a lightweight nnU-Net using the union of labeled images and selected unlabeled images with pseudo labels. Then, we further propose an efficient sliding window strategy based on the prior knowledge of the abdomen to reduce the number of inference windows. Furthermore, we also rewrite the implementation code of the time-consuming part in nnU-Net such as crop and resample.

Our main contributions are summarized as follows:

- We design a pseudo labeling framework based on nnU-Net that can generate high-quality pseudo labels and make inference efficiently simultaneously.
- We propose an image-level pseudo label selection method based on the stability of the pseudo labels during different re-training iterations. Models trained using our selected pseudo labels perform better.
- We propose an efficient sliding-window inference strategy by considering the prior knowledge of the abdominal organ volume. This strategy can greatly reduce the number of inference windows.
- We optimize the time-consuming parts of the code in nnU-Net such as crop and resample.

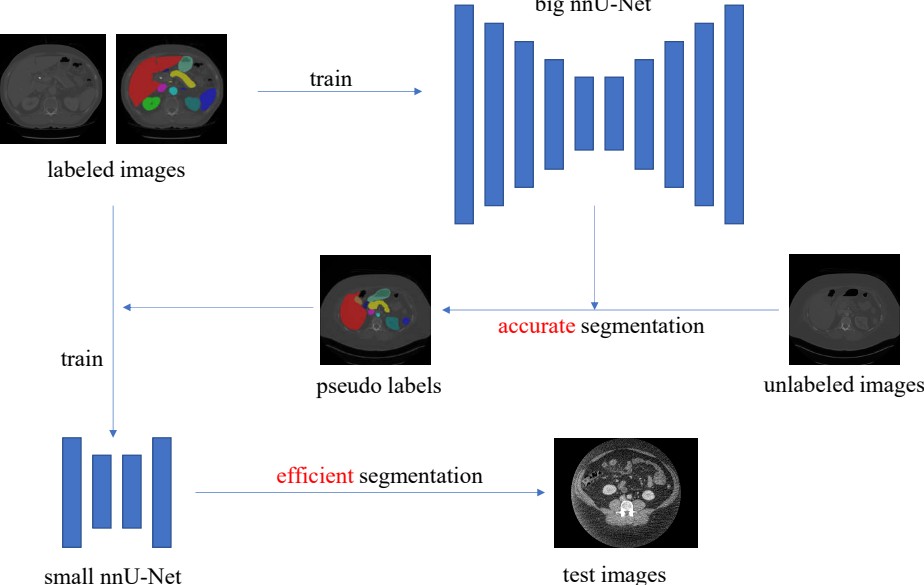

**Fig. 1.** Overview of our proposed framework.

**Table 1.** Comparison of different segmentation strategies. The first two rows evaluate model designs and the remaining rows are for inference strategies. The order of axes of input patch size and spacing is (z,y,x).

| Settings | Default | Accurate | Efficient |
|---|---|---|---|
| channels in the first stage | 32 | 32 | 16 |
| convolution number per stage | 2 | 3 | 2 |
| downsampling times | 5 | 5 | 4 |
| input patch size | (40, 224, 192) | (48, 224, 224) | (32, 128, 192) |
| input spacing | (2.5, 0.8, 0.8) | (2.5, 0.8, 0.8) | (4.0, 1.2, 1.2) |
| test time augmentation | yes | yes | no |

## 2   Method

As illustrated in Figure 1, our framework contains two 3D nnU-Net to achieve high-quality pseudo labeling and efficient inference respectively.

### 2.1   Accurate Segmentation vs Efficient Segmentation

As revealed by EfficientNet[10], deeper and wider networks trained with higher resolution images always have better performance but also cost more computational resources. The default 3D nnU-Net prefers to keep the original resolution of images for better accuracy and resample the spacings of all images to the median spacings of the dataset (10th percentile of the spacings for anisotropic axis). However, the default nnU-Net also makes a compromise on the size of the network and input patch to make the network trainable within 10GB GPU memory.

Based on the default setting of nnU-Net, we design a set of accurate but resource-intensive settings and a set of efficient settings. In the accurate settings, we use a bigger model and a bigger input patch size. In the efficient setting, we not only use a smaller model and smaller input patch size but also resample images to larger spacing. That also means the input images are with lower resolutions for efficient settings. Test time augmentation is applied in the default nnU-Net and our accurate setting, but we do not use it in our efficient setting as it will cost about $8\times$ inference time. The detailed configurations and the comparison with default nnU-Net are listed in Table 1.

For image prepossessing, both of our accurate settings and efficient settings follow the default nnU-Net that clips CT images to 0.5 and 99.5 percentiles of foreground voxels and normalizes images by subtracting the mean then divides by the standard deviation calculated on all images. We do not conduct any postprocessing in our settings.

### 2.2   Iterative Pseudo Labeling by Accurate Segmentation

We adopt pseudo labeling, a simple but effective method, to leverage the unlabeled data for training model. Considering the unsatisfactory performance of

the efficient segmentation strategy mentioned above, which may degrade the quality of pseudo labels, we use the accurate segmentation strategy to generate high-quality pseudo labels for the efficient segmentation strategy.

**Simple Pseudo Labeling Scheme** Our pseudo labeling strategy includes the following steps:

1. Train 5 big nnU-Net models by 5-fold cross-validation on the labeled data.
2. Predict one-hot hard pseudo labels on unlabeled data using our designed accurate inference setting with a 5-fold ensemble of big nnU-Net.
3. Iterative re-train a big nnU-Net on the union of labeled data and unlabeled data with pseudo labels and then generate new one-hot hard pseudo labels for the next round.
4. Select pseudo labels based on the stability of pseudo labels during different training rounds.
5. Train a small nnU-Net on the union of labeled data and selected unlabeled data with pseudo labels for final evaluation.

Here we use the summation between Dice loss and cross-entropy loss because compound loss functions have been proven to be robust in various medical image segmentation tasks [6].

**Pseudo Label Selection** As the trained big nnU-Net may not perform well in all the unlabeled images, some unreliable pseudo labels may harm the training of small nnU-Net. We design a simple method to filter the unreliable pseudo labels based on the stability during different training iterations. We assume that the generated pseudo labels should be stable during iterative training. If some pseudo labels vary greatly in different iterations, it indicates that the model is very uncertain about these pseudo labels and we should not use them for training. We calculate the uncertainty of pseudo labels using the following equation:

$$u = \frac{1}{K-1} \sum_{i=2}^{K} \frac{SUM(y_i \neq y_{i-1})}{SUM(y_i > 0)} \tag{1}$$

where $u$ is the uncertainty and $K$ is the total number of iterations, $y_i$ is the pseudo label generated in iteration $i$.

### 2.3   Efficient Sliding Window Inference

Due to the high resolution of volumetric medical images, nnU-Net adopts the sliding-window strategy for inference. In this strategy, the total inference time depends on the number of windows and the inference time per window. Given input size $(x, y, z)$, window size $(p_x, p_y, p_z)$ and inference step size $s$, the number of sliding window $N$ can be calculated as below:

$$N = \lceil \frac{x - p_x}{s * p_x} \rceil * \lceil \frac{y - p_y}{s * p_y} \rceil * \lceil \frac{z - p_z}{s * p_z} \rceil \tag{2}$$

where $s \in (0, 1]$ and $\lceil \cdot \rceil$ means round up operation.

**Lower Resource Consumption For Each Window** In our efficient inference setting, we use a small model and small patch size as in Table 1 to accelerate the inference speed for each window and also reduce the GPU memory.

**Reduce Total Number of Sliding Window** The default window sliding strategy designs steps for axis x, y, and z separately and uses three layers of for loop to traverse the whole image. However, the abdominal area occupies a small percentage of the entire image, especially in whole-body CT images. With prior knowledge of human anatomy, the region of abdominal organs is expected to have a limited volume and locate in the middle of each transverse section. So we propose to use $3 \times 3$ windows for each transverse section with 50% overlapping. In addition, we first do inference in the middle window, if this window has no foreground area, we can skip surrounding windows.

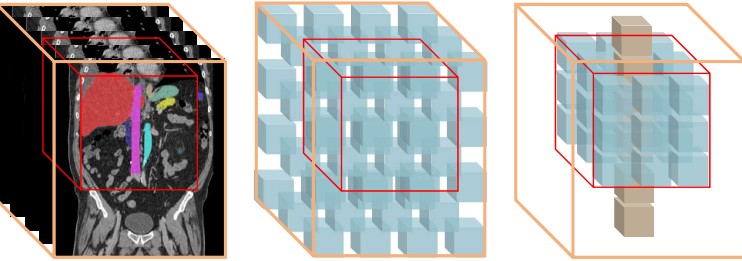

**Fig. 2.** Illustration of our proposed efficient sliding window strategy and comparison with nnU-Net. The red box indicates the region of interest for abdominal organs. The middle figure is the original sliding window strategy used in nnU-Net. The right figure is our proposed strategy that uses the middle window (brown) for every transverse section first to determine whether to do inference for surrounding windows (blue).

## 3    Experiments

### 3.1    Dataset and evaluation measures

The FLARE2022 dataset is collected from more than 20 medical groups under the license permission, including MSD [9], KiTS [2,3], AbdomenCT-1K [8], and TCIA [1]. The training set includes 50 labeled CT scans with pancreas disease and 2000 unlabelled CT scans with liver, kidney, spleen, or pancreas diseases. The validation set includes 50 CT scans with liver, kidney, spleen, or pancreas diseases. The testing set includes 200 CT scans where 100 cases have liver, kidney, spleen, or pancreas diseases and the other 100 cases have uterine corpus endometrial, urothelial bladder, stomach, sarcomas, or ovarian diseases. All the CT scans only have image information and the center information is not available.

The evaluation measures consist of two accuracy measures: Dice Similarity Coefficient (DSC) and Normalized Surface Dice (NSD), and three running efficiency measures: running time, area under GPU memory-time curve, and area under CPU utilization-time curve. All the measures will be used to compute the ranking. Moreover, the GPU memory consumption has a 2 GB tolerance.

### 3.2   Implementation details

The development environments and requirements are presented in Table 2. The training protocols of big nnU-Net and small nnU-Net are listed in Table 3 and  4 respectively. We adopt data augmentation of additive brightness, gamma, rotation, scaling, and elastic deformation on the fly during training. It is noticeable that we use mirror data augmentation for the big model but abandons it for the small model as the small model does not do test time augmentation (TTA) of flipping during inference.

**Table 2.** Development environments and requirements.

| | |
|---|---|
| System version | CentOS Linux release 7.6.1810 |
| CPU | Dual AMD Rome 7742@3.4GHz |
| RAM | 32×32GB; 3200MT/s |
| GPU (number and type) | 8x NVIDIA A100 80GB Tensor Core GPUs |
| CUDA version | 11.2 |
| Programming language | Python 3.8.0 |
| Deep learning framework | Pytorch (Torch 1.10.1) |
| Specific dependencies | nnU-Net 1.7.0 |
| Code | https://github.com/Ziyan-Huang/FLARE22 |

**Table 3.** Training protocols for big nnU-Net.

| | |
|---|---|
| Network initialization | "He" normal initialization |
| Batch size | 2 |
| Patch size | 48×224×224 |
| Total epochs | 1000 |
| Optimizer | SGD with nesterov momentum ($\mu = 0.99$) |
| Initial learning rate (lr) | 0.01 |
| Lr schedule | Poly learning rate policy: $(1 - epoch/1000)^{0.9}$ |
| Training time | 24 hours |
| Loss function | Dice loss and cross entropy loss |
| Number of model parameters | 82M |
| Number of flops | 776G |
| $CO_2$eq | 34.01 Kg |

**Table 4.** Training protocols for small nnU-Net.

| | |
|---|---|
| Network initialization | "He" normal initialization |
| Batch size | 2 |
| Patch size | $32 \times 128 \times 192$ |
| Total epochs | 1500 |
| Optimizer | SGD with nesterov momentum ($\mu = 0.99$) |
| Initial learning rate (lr) | 0.01 |
| Lr schedule | Poly learning rate policy: $(1 - epoch/1500)^{0.9}$ |
| Training time | 12 hours |
| Loss function | Dice loss and cross entropy loss |
| Number of model parameters | 5.4M |
| Number of flops | 136G |
| $CO_2$eq | 11.08 Kg |

## 4    Results and discussion

### 4.1    Quantitative results on validation set

For iterative pseudo labeling, we repeatedly generate pseudo labels for three iterations by using the big nnU-Net and then filter out 76 unreliable pseudo labels in the final iteration. The 76 unreliable pseudo labels is chosen by Equation (1) when the threshold of $\mu$ is set to 0.1. That is, we have 50 labeled images and 1924 images with reliable pseudo labels in the end. We compare the performance of both big nnU-Net and small nnU-Net trained with or without 1924 reliable pseudo labels. We report the results of DSC on the validation leaderboard[1] in Table 5.

**Table 5.** DSC of accurate segmentation and efficient segmentation with and without selected pseudo labels on online validation leaderboard.

| Model | Training Images | Liver | RK | Spleen | Pancreas | Aorta | IVC | RAG | LAG |
|---|---|---|---|---|---|---|---|---|---|
| Big nnU-Net | Labeled Only | 0.9707 | 0.8894 | 0.9228 | 0.8688 | 0.9576 | 0.8950 | 0.8105 | 0.8414 |
| Big nnU-Net | With Pseudo Labels | 0.9802 | 0.9508 | 0.9696 | 0.8965 | 0.9731 | 0.9088 | 0.8481 | 0.8469 |
| Small nnU-Net | Labeled Only | 0.9564 | 0.8655 | 0.9134 | 0.8011 | 0.9292 | 0.8632 | 0.7466 | 0.7005 |
| Small nnU-Net | With Pseudo Labels | 0.9708 | 0.9382 | 0.9537 | 0.8764 | 0.9529 | 0.8909 | 0.7740 | 0.8038 |

| Model | Training Images | Gallbladder | Esophagus | Stomach | Duodenum | LK | Mean |
|---|---|---|---|---|---|---|---|
| Big nnU-Net | Labeled Only | 0.8375 | 0.8696 | 0.9067 | 0.7755 | 0.8903 | 0.8797 |
| Big nnU-Net | With Pseudo Labels | 0.8459 | 0.8894 | 0.9142 | 0.8363 | 0.9233 | 0.9064 |
| Small nnU-Net | Labeled Only | 0.6556 | 0.7931 | 0.8483 | 0.7077 | 0.8485 | 0.8176 |
| Small nnU-Net | With Pseudo Labels | 0.7660 | 0.8653 | 0.8949 | 0.8052 | 0.9127 | 0.8773 |

---

[1] https://flare22.grand-challenge.org/evaluation/challenge/leaderboard/

As shown in Table 5, training models with labeled data and selected data with pseudo labels can improve models' performance compared to training models with only labeled data. Moreover, the improvement is more significant for small models with an efficient inference strategy.

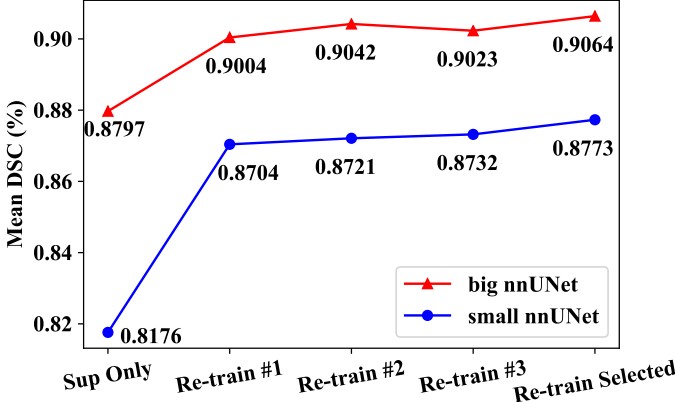

**Fig. 3.** Effectiveness of iterative training and label selection

We examine the effectiveness of iterative training and model selection in Figure 3. We can see that retraining models for more iteration can help improve performance but the performance gain is gradually flattening. We also observe that filtering out some noisy labels can further improve the models' performance.

### 4.2   Qualitative results on validation set

Figure 4 shows 4 representative segmentation results of our small nnU-Net trained on 50 labeled data and 1924 selected pseudo labels for final submission. For Case #21 and Case #35, the network successfully identifies all organs with high accuracy. For Case #42 and Case # 48, it is easy to see that some under-segmentation and over-segmentation errors occurred. We argue that this is due to the small nnU-Net lack of reprehensibility and images after resampling to low resolution lose some important details.

### 4.3   Segmentation efficiency results

We build our small nnU-Net with an efficient inference strategy as a docker image for final submission. In Table 6, we report the efficiency evaluation results on our personal computer with 32 GB RAM, CPU i7-8700 and GPU 1070 using the official evaluation code [2].

---

[2] https://github.com/JunMa11/FLARE/tree/main/FLARE22/Evaluation

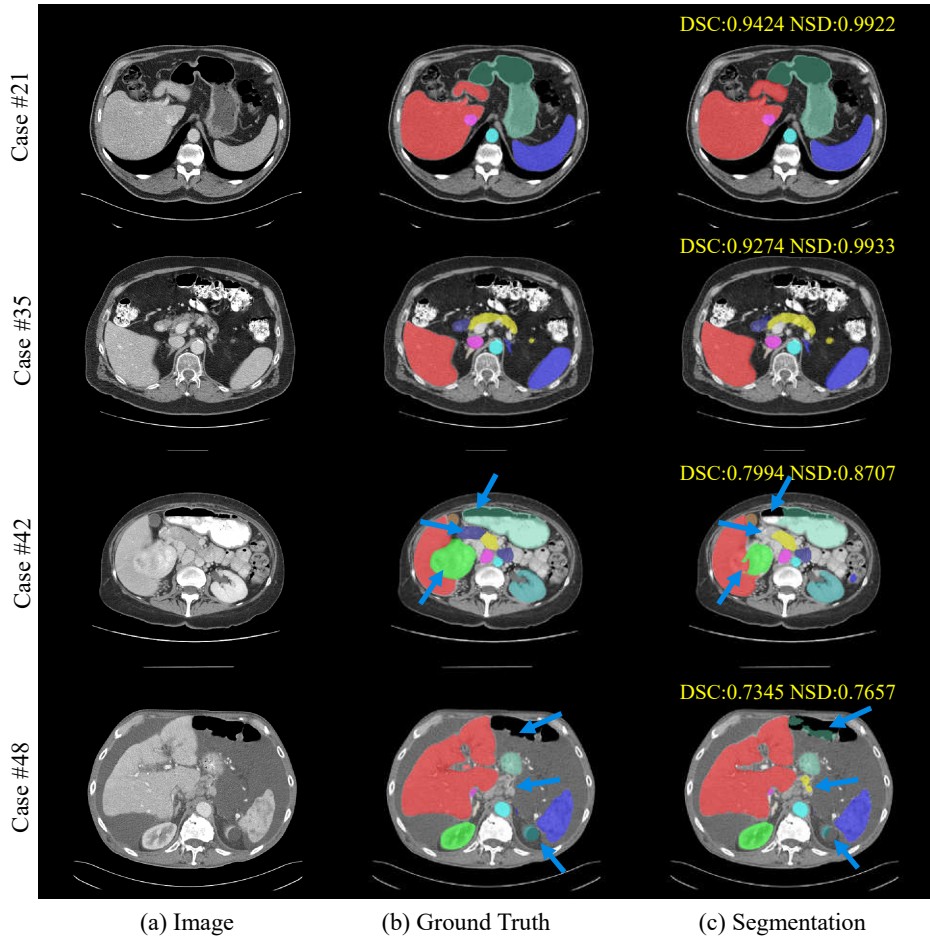

**Fig. 4.** Qualitative results of our small nnU-Net on two easy cases (Case #21 and Case #35) and two hard cases (Case #42 and Case #48).

**Table 6.** Efficiency evaluation results of our submitted docker. All metrics reported are the average values on 50 validation cases

| Time | GPU memory | AUC GPU Time | CPU Utilization | AUC CPU Time |
|------|-----------|--------------|-----------------|--------------|
| 12.8s | 1762MiB | 15990 | 71.8% | 242 |

### 4.4   Results on final testing set

Our method wins the championship among 47 submissions on the final testing set. Table 7 and Table 8 shows the detail evaluation metrics of our method on final testing set.

**Table 7.** Testing results of our proposed method. All metrics reported are the average values on 200 testing cases.

| DSC | NSD | Time | AUC GPU Time | AUC CPU Time |
|---|---|---|---|---|
| 0.8864 | 0.9465 | 14.59s | 14307 | 295 |

**Table 8.** Evaluation metrics of average±standard deviation of DSC and NSD per substructure on 200 testing cases.

| Substructure | Mean DSC | Mean NSD |
|---|---|---|
| Liver | 0.9743±0.0110 | 0.9863±0.0243 |
| Right Kidney | 0.9466±0.1179 | 0.9700±0.1161 |
| Spleen | 0.9432±0.1288 | 0.9624±0.1370 |
| Pancreas | 0.8528±0.1042 | 0.9537±0.1009 |
| Aorta | 0.9559±0.0239 | 0.9884±0.0289 |
| Inferior Vena Cava | 0.9040±0.0619 | 0.9224±0.0688 |
| Right Adrenal Gland | 0.8280±0.0941 | 0.9568±0.1020 |
| Left Adrenal Gland | 0.8286±0.0929 | 0.9591±0.0830 |
| Gallbladder | 0.8340±0.2548 | 0.8486±0.2631 |
| Esophagus | 0.8122±0.1174 | 0.9176±0.1183 |
| Stomach | 0.9237±0.0746 | 0.9609±0.0757 |
| Duodenum | 0.7904±0.1325 | 0.9236±0.1106 |
| Left Kidney | 0.9291±0.1396 | 0.9539±0.1397 |

It is noticeable that our method achieves very good performance in terms of NSD. We argue that the sliding window inference strategy plays an important role in boundary segmentation.

### 4.5   Limitation and future work

Pseudo labeling is a simple and conventional method for semi-supervised learning, but the pseudo label can still be noisy even after the uncertainty-based pseudo label selection. We will refer to the updated research progress to improve the quality of pseudo labels in our future work.

## 5   Conclusion

In this paper, we design a framework based on nnU-Net to use the unlabeled data for training and make inference efficiently. We believe that our proposed framework can serve as a good baseline for semi-supervised learning and efficient inference for medical image segmentation.

**Acknowledgements** The authors of this paper declare that the segmentation method they implemented for participation in the FLARE 2022 challenge has not used any pre-trained models nor additional datasets other than those provided by the organizers. The proposed solution is fully automatic without any manual intervention.

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
