# OpenReview forum: "Revisiting nnU-Net for Iterative Pseudo Labeling and Efficient Sliding Window Inference"
_MICCAI.org/2022/Challenge/FLARE_

### Official Review · Reviewer_9gGK · 2022-09-16
**Good redesign of classical nnunet**

**Rating:** 10
**Confidence:** 3

**Review:**

Strength: The trade-off between model efficiency and accuracy is quite reasonable. The use of nnUnet ensures the generalizability of the model. And the inference speed is guaranteed with a low number of model parameters. Great illustration.

Weakness: No obvious logical error found in the paper except for some minor grammar mistakes. Good job.

---

> ### Author Response · Authors · 2022-10-09
> **Response to Reviewer1**
>
> We thank the reviewers for acknowledging our work. We agree that develop medical image segmentation method based on nnU-Net is general and important, as nnU-Net is currently the most important baseline in this domain. We hope our method can help other researchers to develop semi-supervised method in nnU-Net framework or accelarate the inference in nnU-Net in the future.

---

### Official Review · Reviewer_XARh · 2022-09-16
**Iterative pseudo labeling with nnUNet**

**Rating:** 8
**Confidence:** 4

**Review:**

The authors propose an iterative pseudo-labelling process based on nnU-Net to leverage the unlabeled data. Two nnU-Net models are developed:
a high-performance, high-resource nnU-Net for iterative pseudo label generation
a lower-performing, lower-resource nnU-Net for efficient inference
The authors also change the sliding-window inference approach of nnU-Net to make it more efficient.

Pros:
. A well-written paper, simple method with a thorough description
. Good performance on the validation set with both the big and small networks.
. Reproducible and open-source code.

Cons:
. No reasoning or evidence is shown for the design choices regarding the input patch size and spacing for the big/small nnU-Nets
. There is no ablation study to show the individual influence of every alteration made to nnU-Net for optimizing resource consumption. Namely, the authors changed the architecture, increased the downsampling, disabled test-time augmentations, reduced the patch overlap to 50%, and introduced a different sliding-window approach. Based on the presented results, it is impossible to say if the proposed sliding window approach makes a significant difference in resource consumption and performance.
. For the sliding window approach, the authors define a region of interest for abdominal organs but do not specify how this region is obtained.

---

> ### Author Response · Authors · 2022-10-09
> **Response to Reviewer2**
>
> We thank the reviewer for useful comments.
>
> > No reasoning or evidence is shown for the design choices regarding the input patch size and spacing for the big/small nnU-Nets .
>
>
> We agree that the design of input patch size and spacing for big/samll nnU-Nets is not so convincing and there is still room to improve the design.
> For big nnU-Net, the input patch size and spacing design is roughly the same as the default setting and we just keep the patch size in x axis and y axis the same. We believe the default value is good enough.
> For small nnU-Net, we use some prior knowledge of abdomen CT. The input patch size is (32, 128, 192) and the spacing is (4.0, 1.2, 1.2) and the physical size of each patch is (128mm, 153.6mm, 230.4mm). We use 3×3 window for each xy plane and the overlap ratio is 50%, the area in each xy plane is 460.8mm×307.2mm and it can cover the abdomen organs region in almost all people. (The length of the x axis is longer).
>
> > There is no ablation study to show the individual influence of every alteration made to nnU-Net for optimizing resource consumption
>
> We thank the reviewer for pointing out this issue. We agree that ablation study for each individual component is important to recognize the influence of each design. However, these experiments are time-consuming, and we can't finish them quickly. We will consider make an extension of our work and that version will have thorough ablation experiments.
>
> > it is impossible to say if the proposed sliding window approach makes a significant difference in resource consumption and performance.
>
> The improvement of inference speed from our proposed efficient sliding window inference depends on the specific case. For 50% cases in validation set, the number of windows for inference can be roughly halved. For some extreme whole-body CT, the number of windows can be reduced to 10%.
>
> > For the sliding window approach, the authors define a region of interest for abdominal organs but do not specify how this region is obtained.
>
> In fact, we do not define a region of interest for abdominal organs, we just skip some surrounding background region in the CT image by our proposed sliding window strategy. *We first do inference in the middle window, if this window has no foreground area, we can skip surrounding windows.*

---

### Official Review · Reviewer_r4xJ · 2022-09-17

**Rating:** 8
**Confidence:** 4

**Review:**

**Summary:**

This work is a nnU-Net based framework and achieves mDSC of 0.8773 in only about 12 seconds for each case. It's meaningful that this work is several times faster than the nnU-Net while maintaining the same performance as nnU-Net.

**Strengths:**

- Similar or even higher performance compared with nnU-Net but much faster than nnU-Net.
- Simple but efficacious pseudo Labeling scheme for semi-supervised learning.
- Efficient sliding window strategy. It will significantly shorten inference time by skipping windows without foreground area.

**Questions:**

- In this work, the supervised nnU-Net achieves mDSC of 0.8797 and even gets 0.9064 after re-training, as shown in Fig. 3. In my experience, the pure nnU-Net can not achieve this (most works based on nnU-Net achieve mDSC around 0.85~0.87 in the FLARE2022). Thus, my first question is, how does the big nnU-Net achieve such high performance in supervised training? Is it trained on nnU-Net default deployments or with some other tricks not mentioned?
- How much time does the efficient sliding window strategy reduce in the inference stage? How much performance will be dropped by using this sliding window strategy? These are not mentioned in this work.

---

> ### Author Response · Authors · 2022-10-09
> **Response to Reviewer3**
>
> We thank the reviewer for positive comments and valuable concerns.
> > how does the big nnU-Net achieve such high performance in supervised training? Is it trained on nnU-Net default deployments or with some other tricks not mentioned?
>
> Compared to default nnU-Net, the most important modification is that we use a bigger model (we increase convolution number in each stage from 2 to 3). Bigger model always has better performance. Besides, we add the elastic data augmentation compared to default data augmentation, it needs more training time but can get performance gain. We have described these modifications in our paper explicitly.
>
> > How much time does the efficient sliding window strategy reduce in the inference stage? How much performance will be dropped by using this sliding window strategy? These are not mentioned in this work.
>
> The improvement of inference speed from our proposed efficient sliding window inference depends on the specific case. For 50% cases in validation set, the number of windows for inference can be roughly halved. For some extreme whole-body CT, the number of windows can be reduced to 10%. For performance drop, in fact the efficient sliding window strategy cause no performance drop as we only carefully skip some background area.

---

### Official Review · Reviewer_BGXJ · 2022-09-19
**Strong iterative noisy student approach based on nnU-Net, just some minor details missing.**

**Rating:** 9
**Confidence:** 4

**Review:**

Summary:
This paper combines a noisy student approach with nnU-Net. Here, 5 large teacher models are trained on the labeled data, which are then ensembled to generate pseudo labels on the unlabeled dataset. The pseudo labels are re-used to iteratively improve the large teacher models on both the labeled and pseudo-labeled images.
For the final efficient student network, a small 3D U-Net is trained on the union of labeled and filtered pseudo-labeled images. The filtering is based on an image-level consistency of segmentation maps between the different large-model re-training iterations.

Pros:
- Very well written, clear description of the strategy and methodology used.
- Figures support the understanding of the approach in a clear fashion.
- The filtering approach is very intuitive and a nice way to filter for stability.
- The efficient sliding window approach is a good approach to save inference time.

Missing information:
- How exactly is the filtering done (uncertainty threshold u)? The 76 filtered unreliable pseudo labels seem a bit ad-hoc.
- A comparison to the big nnU-Net in terms of inference time, GPU memory etc. would have been interesting.

Problems:
- The resolution of Figure 4 is a bit low, which makes it hard to inspect segmentation details (minor).
- Typo in section 4.3: dcoker instead of docker (minor).

---

> ### Author Response · Authors · 2022-10-09
> **Response to Reviewer4**
>
> We thank the reviewer for detailed and useful comments.
> > How exactly is the filtering done (uncertainty threshold u)? The 76 filtered unreliable pseudo labels seem a bit ad-hoc.
>
> Thank you for pointing out this important issue. We empirically set the threshold $u$ to 0.1. We have added it in section 4.1. As we find training without filtering is good enough, so we do not want to filter too many pseudo labels. To be honest, we do not try other threshold since there is no time left for us to adjust it because of the challenge DDL.
>
> > A comparison to the big nnU-Net in terms of inference time, GPU memory etc. would have been interesting.
>
> Thank you for your suggestion. We agree that the comparison between big nnU-Net and small nnU-Net in terms of efficiency is important. We test the small nnU-Net docker on my personal computer with GTX 1070 GPU and 32 GB RAM, however the big nnU-Net cannot be tested on it.
>
> > The resolution of Figure 4 is a bit low, which makes it hard to inspect segmentation details
>
> We have redrawn the Figure4 for higher resolution, we also add some arrows and the DSC, NSD value on it. We hope it is clear enough now.
>
> > Typo in section 4.3: dcoker instead of docker.
>
> We have modified it.

---

### Official Review · Reviewer_kZte · 2022-09-21
**The resource-intensive uu U-Net model achieved a mean DSC of 0.9064 on 13 abdominal organ segmentation tasks and ranked first on the validation leaderboard. However, still have some issues.**

**Rating:** 7
**Confidence:** 4

**Review:**

In this manuscript, the authors designed a co-training framework consisting of two strategies to generate high-quality pseudo labels and make efficient inferences, respectively. The resource-intensive uu U-Net model achieved a mean DSC of 0.9064 on 13 abdominal organ segmentation tasks and ranked first on the validation leaderboard.

Strengths: The model achieved the top score on the leaderboard in terms of mean DSC.

Weaknesses and queries for authors:

1: There is no ablation study that proves the effectiveness of the model. Suggested to add.

2: No NSD was reported in the manuscript. Suggested to add.

3: The paper structure could be modified and suggested to restructure; things are added in unarranged manners or have missing components/connections.

3: While training the lightweight nn U-Net, do you use the sliding window? If so, then how could it be beneficial for faster inference?

4: Figure 1 should be modified entirely, which is more apparent concerning a clear explanation for each component.

5: Table 5 is split; it is recommended to make it unified for clarity.

6: Figure 4 has captions at the bottom rather than on the top.

7: Throughout the manuscript, there are many grammatical errors and typos. Therefore, it is suggested that the whole manuscript could be checked for such mistakes and should be fixed, such as,

- nnUUNet should be nnU-Net

- 3d should be 3D

---

> ### Author Response · Authors · 2022-10-09
> **Response to Reviewer5**
>
> We thank the reviewer for constructive comments.
> > 1: There is no ablation study that proves the effectiveness of the model. Suggested to add.
>
> As the effectiveness of the model, the reviewer might have overlooked Table 5 and Figure 3. Also, the top-1 performance on validation leaderboard may prove the effectiveness of the model.
>
> > 2: No NSD was reported in the manuscript. Suggested to add.
>
> We agree with the reviewer. We have reported the NSD value on test set in Sec 4.4. In Table 5, we compare model performance on the online validation leaderboard. As only DSC value on the leaderboard, so we cannot report NSD value.
>
> > 3: While training the lightweight nn U-Net, do you use the sliding window? If so, then how could it be beneficial for faster inference?
>
> The training in nnU-Net framework is sampling-based, that means it samples some patch for each training iteration not the sliding window. The sliding window is only used for inference in nnU-Net, and we accelerate it by our proposed efficient sliding window inference.
>
> > 4: Figure 1 should be modified entirely, which is more apparent concerning a clear explanation for each component.
>
> Thank you for the comment, but we cannot fully agree with the comment. We have explained each component in the Figure simply. As each component is simple enough (as labeled image, pseudo label, big nnU-Net), we feel that no further explanation is necessary.
>
> > 5: Table 5 is split; it is recommended to make it unified for clarity.
>
> The guideline from challenge organizers is "*You can present the results in multiple rows or vertically display it.*" So, we think it is acceptable.
>
> > 6: Figure 4 has captions at the bottom rather than on the top.
>
> The caption of Figure4 is on the bottom already. We use the official $latex$ template.
>
> > 7: Throughout the manuscript, there are many grammatical errors and typos. Therefore, it is suggested that the whole manuscript could be checked for such mistakes and should be fixed, such as, nnUUNet should be nnU-Net, 3d should be 3D.
>
> We thank the good suggestion. We have modified nnUNet to nnU-Net and 3d to 3D. Also, we have checked the manuscript thoroughly with Grammarly.

---

### Meta-Review · Program_Chairs · 2022-09-28

**Recommendation:** Minor Revision
**Confidence:** 5

**Metareview:**

Nice paper. Please address the reviewers' comments in the revised manuscript.